# The burden of skin disease and eye disease due to onchocerciasis in countries formerly under the African Programme for Onchocerciasis Control mandate for 1990, 2020, and 2030

**Natalie V. S. Vinkeles Melchers**[1]*, **Wilma A. Stolk**[1], **Welmoed van Loon**[1,2], **Belén Pedrique**[3], **Roel Bakker**[1], **Michele E. Murdoch**[4], **Sake J. de Vlas**[1], **Luc E. Coffeng**[1]*

1 Department of Public Health, Erasmus MC, University Medical Center Rotterdam, Rotterdam, The Netherlands, 2 Institute of Tropical Medicine and International Health, Charité-Universitätsmedizin, Berlin, Germany, 3 Drugs for Neglected Diseases *initiative* (DND*i*), Geneva, Switzerland, 4 Department of Dermatology, West Herts Hospitals NHS Trust, Watford General Hospital, Watford, Hertfordshire, United Kingdom

* n.vinkelesmelchers@erasmusmc.nl (NVSVM); l.coffeng@erasmusmc.nl (LEC)

## Abstract

### Background

Onchocerciasis ("river blindness") can cause severe morbidity, including vision loss and various skin manifestations, and is targeted for elimination using ivermectin mass drug administration (MDA). We calculated the number of people with *Onchocerca volvulus* infection and onchocercal skin and eye disease as well as disability-adjusted life years (DALYs) lost from 1990 through to 2030 in areas formerly covered by the African Programme for Onchocerciasis Control.

### Methods

Per MDA implementation unit, we collated data on the pre-control distribution of microfilariae (mf) prevalence and the history of control. Next, we predicted trends in infection and morbidity over time using the ONCHOSIM simulation model. DALY estimates were calculated using disability weights from the Global Burden of Disease Study.

### Results

In 1990, prior to MDA implementation, the total population at risk was 79.8 million with 26.0 million (32.5%) mf-positive individuals, of whom 17.5 million (21.9%) had some form of onchocercal skin or eye disease (2.5 million DALYs lost). By 2030, the total population was predicted to increase to 236.1 million, while the number of mf-positive cases (about 6.8 million, 2.9%), people with skin or eye morbidity (4.2 million, 1.8%), and DALYs lost (0.7 million) were predicted to decline.

**Data Availability Statement:** All relevant data are within the manuscript and its Supporting Information files.

**Funding:** NVSVM, WAS, WvL, BP, and LEC received funding from the United States Agency for International Development (USAID, www.usaid.gov) through the Drugs for Neglected Diseases initiative (DNDi, www.dndi.org) (ref no. AID-OAA-G14-00010). The contents are the responsibility of the authors and do not necessarily reflect the views of USAID or the United States Government. For its overall mission, DNDi receives support from UK aid, UK (www.ukaiddirect.org; MOU 2013-2018 and MOU 2017-2021); Médecins sans Frontières (MSF, www.msf.org; MOU 2014-2018 and MOU 2019-2023); and the Swiss Agency for Development and Cooperation, Switzerland (SDC, www.eda.admin.ch/sdc; contract no. 81017718 and no. 81050394). WAS, SJdV, and LEC gratefully acknowledge funding of the NTD Modelling Consortium by the Bill & Melinda Gates Foundation (www.gatesfoundation.org; grant OPP1184344). LEC further acknowledges funding from the Dutch Research Council (NWO, www.nwo.nl; grant 016.Veni.178.023). The funders had no role in study design, data collection and analysis, decision to publish, or preparation of the manuscript.

**Competing interests:** The authors have declared that no competing interests exist.

## Conclusions

MDA has had a remarkable impact on the onchocerciasis burden in countries previously under the APOC mandate. In the few countries where we predict continued transmission between now and 2030, intensified MDA could be combined with local vector control efforts, or the introduction of new drugs for mopping up residual cases of infection and morbidity.

## Author summary

Onchocerciasis, also known as river blindness, is a neglected tropical disease caused by a parasitic worm transmitted through the bite of an infected blackfly. Onchocerciasis is, or used to be, endemic in many West, Central, and East African countries. Mass drug administration (MDA) with ivermectin and vector control have been used to prevent the spread of infection and effectively control onchocerciasis as a public health problem. In Central and East Africa, this was done under the mandate of the African Programme for Onchocerciasis Control (APOC). In 2012, the World Health Organization targeted onchocerciasis for control and elimination. Here, we assess the impact of MDA with ivermectin on the prevalence of onchocercal morbidity in 1990 (pre-control), 2020, and 2030, and calculate the associated burden of disease in terms of disability-adjusted life years (DALYs), a composite measure of life years lost and years lived with disability. By 2030, we expect that 691 thousand DALYs will be lost in countries formerly covered by APOC. This burden is due to onchocercal skin disease in about 3.8 million people and onchocercal eye disease in about 384 thousand people, with most of this burden being concentrated in only a few countries.

## Introduction

Human onchocerciasis, also known as river blindness, is a parasitic disease caused by *Onchocerca volvulus* that is transmitted through the bite of an infected blackfly (genus *Simulium*). Onchocerciasis has been associated with a high impact on health and socioeconomic status due to blindness and stigmatising scaly, itchy skin manifestations [1]. Before the initiation of mass drug administration (MDA), about 32 million people were estimated to be infected with *O. volvulus* in countries under the former mandate of the African Programme for Onchocerciasis Control (APOC), which ran from 1995 till 2015 [2]. Another 7.6 million people were estimated to be infected in areas previously under the mandate of the Onchocerciasis Control Programme (OCP, 1974–2002) in West-Africa, prior to the initiation of interventions [3]. In 2019, an estimated 217.2 million people in 30 countries were in need of preventive therapy for onchocerciasis [4], most of whom live in sub-Saharan Africa. Thanks to long-term and large-scale onchocerciasis interventions in Africa, 1.2 million people no longer require MDA [4] and several other regions may be close to elimination [5].

Onchocerciasis is now targeted for elimination [6], and great progress has been achieved in reducing *O. volvulus* infection to low levels with MDA [5]. However, with the current focus being mostly on infection levels and interruption of transmission, relatively little attention has been paid to the expected impact of MDA on the prevalence of clinical manifestations and overall disease burden. This is relevant because there are likely to be residual infections and disease burden even after elimination targets are met or by the time they should be met. In

areas where the targets are met in time and MDA is scaled down, some form of clinical treatment may be needed to address residual cases of infection and disease (e.g. macrofilaricidal drugs or drugs inducing permanent sterilisation of adult female worms). For areas where MDA was implemented relatively recently or less successfully, additional interventions may be needed to speed up progress towards elimination, e.g. intensified MDA efforts, use of moxidectin (longer suppression of skin mf), additional vector control, or new macrofilaricidal or worm-sterilising drug treatments. To understand the potential need for treatments with a new drug to address the residual burden of onchocerciasis, it is necessary to quantify to what extent this burden is affected by MDA with ivermectin. It is important to note that onchocercal morbidity includes a wide range of clinical manifestations, some of which will (partially) persist even after clearance of infection (chronic morbidity, e.g. vision loss and skin depigmentation) and are therefore not, or only marginally, amenable to anti-parasitic treatment.

Here, we estimate the number of infections, number of cases with clinical manifestations, and the disease burden of onchocerciasis in terms of case numbers and disability-adjusted life years (DALYs) lost in 1990, 2020, and 2030 for countries formerly under the APOC mandate. We include a wide spectrum of onchocercal skin (OSD) and eye disease (OED): severe itch, reactive skin disease, skin depigmentation (leopard skin), skin atrophy, hanging groin, subcutaneous onchocercal nodules, and vision loss (visual impairment and blindness). We predict trends over time in the prevalence of infection, OSD, and OED based on data on the pre-control distribution of infection and the history of MDA, and a newly developed version of the mathematical model ONCHOSIM that captures the dynamic association between infection, treatment, and morbidity [7]

## Methods

### General approach

To estimate the burden of onchocerciasis across countries previously under the APOC mandate, we used published pre-control maps of onchocerciasis prevalence covering all APOC countries. We then coupled the pre-control infection prevalence per APOC project (i.e. implementation unit for MDA) with rural population density data at baseline. We used the mathematical model ONCHOSIM [2,8–10] to predict the prevalence of infection and morbidity over time, taking account of the pre-control distribution of infection levels, bioclimate (forest vs. savanna parasite strain), and history of control (MDA, vector control) of each APOC project. Development of morbidity was modelled using a new version of ONCHOSIM that captures the dynamic association between infection, treatment, and morbidity [7]. DALYs lost due to onchocerciasis were quantified by calculating the number of years lived with disability (YLDs) and years of life lost (YLL) due to blindness-related excess mortality. A detailed description of the methods applied is provided in **S1 Text**; a high-level overview is provided below.

### Pre-control infection level and population size

We used a previously published 1x1 km$^2$ resolution raster map of the pre-control prevalence of onchocercal palpable nodules across 18 former APOC countries (i.e. Angola, Burundi, Cameroon, Central African Republic, Chad, Congo, Democratic Republic of Congo, Equatorial Guinea, Ethiopia, Gabon, Liberia, Malawi, Mozambique, Nigeria, South Sudan, Sudan, Tanzania, Uganda), based on Rapid Epidemiological Mapping of Onchocerciasis (REMO) [11,12]. APOC covered 20 countries in Africa. In Rwanda and Kenya, the prevalence of palpable nodules was virtually zero during REMO surveys [11,12]. As a result, these countries were considered non-endemic for onchocerciasis and we therefore did not include them in our analysis. We converted the pre-control prevalence of palpable nodules in adult males into pre-

control microfilariae (mf) prevalence in the population aged ≥5 years, using a published statistical association [13]. We linked each raster cell to an MDA implementation unit, called a "project" from here on, and categorised pixels into six endemicity categories (see **S1 Text**, **Tables A** and **B**). Further, for each project we collated population size estimates from the APOC census database [14,15] and divided the population over the six endemicity levels based on the pixel distribution over the various endemicity levels. See **S1 Text**, sections 2.1 and 2.2 for more details.

## History of control

For each project, we obtained information on the project-specific history of control and expected future treatment scenarios (i.e. MDA start year, achieved coverage, treatment frequency, as well as vector control). **S1 Text**, section 2.3 provides the relevant details of the data on the history of control per project and assumptions on future MDA scenarios. Briefly, nearly all onchocerciasis hyperendemic and mesoendemic areas had started annual or sometimes biannual MDA by 2013 (up to which date we were able to collate information on the history of control [15]). For the few projects that had not yet started MDA by 2013, we consulted the ESPEN Portal [16] to verify the (expected) MDA start year, and adjusted the APOC treatment database where necessary (up to 2017). Hypoendemic areas that are potentially co-endemic for loiasis were assumed not to start MDA before 2025 due to the high risk of serious adverse events related to ivermectin-intake in *Loa loa*-infected individuals, in line with previous work [14]. We accounted for MDA disruptions due to security issues in countries in civil war (CAR, Liberia, South Sudan) and due to the COVID-19 pandemic. An overview of the history of control used in our analysis can be found in **S2 Text**.

## Mathematical modelling

We predict the prevalence of *O. volvulus* infection and clinical manifestations per APOC project from 1990 through to 2030, using a newly developed version of the mathematical model ONCHOSIM that captures the dynamic association between infection, treatment, and morbidity [7]. A brief description of ONCHOSIM and its generic morbidity module is provided in **S1 Text**, sections 3.1 and 3.2. The model was used to predict the mf prevalence in the total population and in individuals aged five years and older, as well as the prevalence of infection with at least one female adult worm in the total population. The model further predicts the prevalence of the following clinical manifestations: severe itch (defined as troublesome itch with insomnia), reactive skin disease (RSD), palpable nodules (which may lead to stigmatisation and shame [17,18]), hanging groin, skin atrophy, depigmentation (two levels of severity: mild and severe), and vision loss (visual impairment and blindness).

Per APOC project (N = 158), we calibrated transmission parameters to reproduce the pre-control mean mf prevalence for each of the six endemicity categories. Next, for each endemicity category in each project, we ran 500 repeated stochastic simulations, accounting for the project-specific MDA history and vector control (**S2 Text**), assuming that endemicity categories within a project experienced the same history of control. Model predictions were saved by age (0-<2, ≥2-<5, ≥5-<10, ≥10-<20, ≥20-<30, ≥30-<50, ≥50), sex, project, and endemicity category, and were averaged over repeated simulations.

Because ONCHOSIM simulates a single closed community of humans and does not consider mobility of infected humans and flies between multiple communities, it cannot simulate stable hypoendemic infection levels. It is yet unclear which mechanisms contribute the stabilisation of infection across hypoendemic areas, but several hypotheses have been raised, which may not be mutually exclusive: spill-over from adjacent more highly endemic areas, more

efficient transmission at low levels through additional density-dependent processes in the blackfly [19], high levels of local individual variation in exposure to blackflies, and/or assortative mixing of small high-risk sub-populations [20]. Rather than explicitly modelling hypoendemic areas making assumptions, we calculated the prevalence of infection and morbidity in hypoendemic areas as a ratio of the model-predicted disease prevalence in mesoendemic areas. We used different ratios for calculating the prevalence of palpable nodules, reversible clinical manifestations (i.e. severe itch and RSD), and irreversible clinical manifestations (i.e. depigmentation, hanging groin, and atrophy) in hypoendemic areas based on a meta-analysis of published data (**Fig B** in **S1 Text**) [21,22]. Further information and details are provided in **S1 Text**, section 3.3.

## Burden calculation

The disease burden of onchocerciasis was quantified in terms of DALYs, which are defined as the sum of Years Lived with Disability (YLDs) and Years of Life Lost (YLLs) [23,24]. YLDs were calculated by multiplying the predicted number of prevalent cases with a weight representing the severity level of the condition (disability weight). Disability weights for various subtypes of OSD were defined based on an earlier scheme of disability weights for onchocercal skin morbidity used in the 2010 edition of the Global Burden of Disease (GBD) study [25], as presented in **Table C** in **S1 Text**. YLD estimates were corrected for concurrence of multiple types of skin manifestations, using a multiplicative approach [26–29]. This correction was considered important as the mechanics of how some skin manifestations cause a disease burden are similar (e.g. disfigurement) and because, conceptually, the sum of disability weight for concurrent symptoms should not exceed 1.0. A detailed description of how the correction was applied, including a graphical representation, is described in **S1 Text**, section 4.1. YLLs were calculated for onchocercal blindness only, as blindness is associated with excess mortality [30,31]. **S1 Text**, section 4.2 provides more details about YLL calculations.

## Sensitivity analysis

We assessed the impact of various biological and programmatic assumptions on the estimated number of cases with infection and morbidity by 2030 through univariate sensitivity analyses, including, but not limited to, assumptions about the ratio in morbidity prevalence between hypoendemic and mesoendemic areas, and over-reporting of MDA coverage. **Table E** in **S1 Text** provides a full overview of all sensitivity analyses performed.

## Results

### Number of cases with infection

The total population living in countries previously under the APOC mandate was predicted to increase from 79.8 million in 1990 to 236.1 million people in 2030. Before the initiation of MDA, the prevalence of *O. volvulus* skin mf-positivity among the population (all ages) was estimated to be 32.5% in 1990 (26.0 million cases), and was predicted to decline to 2.9% by 2030 (6.8 million cases) thanks to MDA, which is a reduction in prevalence of 74%. The prevalence of people infected with at least one adult female worm was predicted to decline from 41.2% in 1990 to 6.6% in 2030, which is a 84% reduction (**Table 1**). As expected, by 2030, prevalence of mf and worms will be highest in areas that were very hyperendemic before the start of control (**Table 2**). Further, by 2030 most mf-positive cases will be in the Democratic Republic of Congo (45.4% of all cases), Nigeria (21.6%) and Ethiopia (10.0%). Highest country-level mf prevalence is expected in Gabon (11.9%), where currently no MDA programme is in place

**Table 1. Population at risk, total number of cases infected and with clinical manifestations, and DALYs lost due to onchocerciasis for 1990, 2020, and 2030.** Absolute numbers (N and DALYs) are presented in thousands.

| | 1990 | | | 2020 | | | 2030 | | |
|---|---|---|---|---|---|---|---|---|---|
| | N | % | DALYs | N | % | DALYs | N | % | DALYs |
| Total population at risk | 79,768 | 100 | - | 180,004 | 100 | - | 236,102 | 100 | - |
| Mf-positive | 25,964 | 32.5 | - | 14,092 | 7.8 | - | 6,813 | 2.9 | - |
| Mf-positive (age 5+) | 25,922 | 32.5 | - | 14,084 | 7.8 | - | 6,812 | 2.9 | - |
| Adult female worm infection | 32,825 | 41.2 | - | 27,073 | 15.2 | - | 15,577 | 6.6 | - |
| Palpable nodules | 7,617 | 9.5 | 81 | 3,810 | 2.1 | 40 | 1,412 | 0.6 | 15 |
| Severe itch | 4,115 | 5.2 | 747 | 1,663 | 0.9 | 302 | 542 | 0.2 | 98 |
| Reactive skin disease | 3,089 | 3.9 | 143 | 845 | 0.5 | 39 | 129 | 0.1 | 6 |
| **Total reversible onchocercal skin disease** | **7,205** | **9** | **890** | **2,508** | **1.4** | **341** | **670** | **0.3** | **104** |
| Mild depigmentation | 935 | 1.2 | 10 | 1,117 | 0.6 | 12 | 879 | 0.4 | 9 |
| Severe depigmentation | 1,057 | 1.3 | 68 | 1,181 | 0.7 | 76 | 863 | 0.4 | 56 |
| Atrophy | 16 | <0.05 | <0.5 | 3 | <0.05 | <0.5 | <0.5 | <0.05 | <0.5 |
| Hanging groin | 33 | <0.05 | 13 | 27 | <0.05 | 10 | 15 | <0.05 | 6 |
| **Total irreversible onchocercal skin disease** | **2,041** | **2.6** | **91** | **2,328** | **1.3** | **99** | **1,756** | **0.7** | **71** |
| **Total onchocercal skin disease** | **16,862** | **21.1** | **1,062** | **8,647** | **4.8** | **480** | **3,839** | **1.6** | **190** |
| Visual impairment | 421 | 0.5 | 13 | 426 | 0.2 | 13 | 316 | 0.1 | 10 |
| Blindness | 194 | 0.2 | 1,397 | 126 | 0.1 | 903 | 69 | <0.05 | 491 |
| **Total onchocercal eye disease** | **616** | **0.8** | **1,410** | **552** | **0.3** | **916** | **384** | **0.2** | **501** |
| **Total all manifestations** | **17,478** | **21.9** | **2,472** | **9,198** | **5.1** | **1,397** | **4,223** | **1.8** | **691** |

* *Note*: Total onchocercal skin disease is the sum of palpable nodules, reversible and irreversible skin disease.

due to *Loa loa* co-endemicity, the Republic of Congo (6.1%), and Mozambique (5.7%) (**Table 3**). At the project-level, we predict that mf prevalence will be highest in onchocerciasis-hypoendemic areas that are co-endemic for loiasis with an overall mean mf prevalence of 12.7% (4.3 million mf-positive cases among 33.5 million population at risk of infection in P5-projects in 12 (suspected) *L. loa*-endemic countries) in 2030 (**Table W** in **S3 Text**). There are also some hyperendemic foci with high local mf prevalence (>10%) remaining in the Central African Republic, Democratic Republic of Congo, and South Sudan by 2030. See **S3 Text** for further detailed estimates by year and project, and **S4 Text** for detailed estimates by APOC-project and country, which also presents country-specific MDA coverages.

## Number of cases with onchocercal morbidity

Prior to MDA implementation (1990), 21.9% of the total population at risk experienced clinical manifestations due to onchocerciasis (17.5 million cases). The most common clinical manifestation was the presence of palpable nodules, followed by acute reversible skin conditions (either severe itch or RSD) (**Table 1**). The total pre-control prevalence of OED was predicted to be 0.8% (615.5 thousand cases), mostly due to visual impairment in savanna areas (82.3% of all OED cases in 1990; **Table G** in **S3 Text**).

By 2030, the total number of cases with onchocercal morbidity was predicted to decline to 4.2 million (1.8% of total population at risk), with palpable nodules and depigmentation contributing most cases (**Table 1**). Irreversible symptoms of onchocerciasis and, in particular, skin manifestations will be responsible for the majority of the cases: by 2030 there will be 1.8 million cases of irreversible OSD and 384 thousand cases of OED (some cases will have both). In contrast, reversible symptoms only affect 670 thousand people (some of whom will also have irreversible symptoms) by 2030. Cases of OED will be mostly concentrated in savanna

**Table 2. Population at risk and numbers of cases infected and with clinical manifestations by endemicity level for 2030.** Absolute numbers (N and DALYs) are presented in thousands.

| | Hypoendemic | | | Mesoendemic | | | Hyperendemic | | | Very hyperendemic | | |
|---|---|---|---|---|---|---|---|---|---|---|---|---|
| | N | % | DALYs | N | % | DALYs | N | % | DALYs | N | % | DALYs |
| Total population at risk | 145,846 | 100 | - | 67,483 | 100 | - | 18,053 | 100 | - | 4,721 | 100 | - |
| Mf-positive | 4,299 | 2.9 | - | 1,237 | 1.8 | - | 692 | 3.8 | - | 585 | 12.4 | - |
| Mf-positive (age 5+) | 4,299 | 2.9 | - | 1,237 | 1.8 | - | 691 | 3.8 | - | 585 | 12.4 | - |
| Adult female worm infection | 8,735 | 6.0 | - | 3,326 | 4.9 | - | 2,083 | 11.5 | - | 1,432 | 30.3 | - |
| Palpable nodules | 482 | 0.3 | 5 | 419 | 0.6 | 4 | 281 | 1.6 | 3 | 231 | 4.9 | 2 |
| Severe itch | 353 | 0.2 | 64 | 107 | 0.2 | 19 | 56 | 0.3 | 10 | 26 | 0.5 | 5 |
| Reactive skin disease | 89 | 0.1 | 4 | 23 | <0.05 | 1 | 11 | 0.1 | <0.5 | 6 | 0.1 | <0.5 |
| **Total reversible onchocercal skin disease** | **442** | **0.3** | **68** | **130** | **0.2** | **21** | **67** | **0.4** | **11** | **32** | **0.7** | **5** |
| Mild depigmentation | 366 | 0.3 | 4 | 274 | 0.4 | 3 | 174 | 1 | 2 | 65 | 1.4 | 1 |
| Severe depigmentation | 277 | 0.2 | 18 | 233 | 0.3 | 15 | 238 | 1.3 | 15 | 115 | 2.4 | 7 |
| Atrophy | <0.5 | <0.05 | <0.5 | <0.5 | <0.05 | <0.5 | <0.5 | <0.05 | <0.5 | <0.5 | <0.05 | <0.5 |
| Hanging groin | <0.5 | <0.05 | <0.5 | 1 | <0.05 | <0.5 | 6 | <0.05 | 2 | 8 | 0.2 | 3 |
| Total irreversible onchocercal skin disease | **643** | **0.4** | **22** | **508** | **0.8** | **18** | **418** | **2.3** | **20** | **188** | **4** | **11** |
| **Total onchocercal skin disease*** | **1,566** | **1.1** | **95** | **1,057** | **1.6** | **43** | **765** | **4.2** | **33** | **451** | **9.5** | **18** |
| Visual impairment | 192 | 0.1 | 6 | 69 | 0.1 | 2 | 38 | 0.2 | 1 | 17 | 0.3 | 1 |
| Blindness | 34 | <0.05 | 227 | 10 | <0.05 | 74 | 14 | 0.1 | 102 | 11 | 0.2 | 87 |
| **Total onchocercal eye disease** | **226** | **0.2** | **233** | **80** | **0.1** | **77** | **51** | **0.3** | **103** | **27** | **0.6** | **87** |
| **Total all manifestations** | **1,792** | **1.2** | **329** | **1,137** | **1.7** | **120** | **817** | **4.5** | **137** | **478** | **10.1** | **106** |

\* *Note*: Total onchocercal skin disease is the sum of palpable nodules, reversible and irreversible skin disease.

areas (87.8% of all OED cases). The highest prevalence of onchocercal morbidity will be in very hyperendemic areas with about 10.1% of the population having clinical manifestations, yet 42.4% of all cases with clinical manifestations due to onchocerciasis were predicted to live in hypoendemic areas where most of the population at risk resides (**Table 2**). By 2030, the majority of clinical cases will be living in the Democratic Republic of Congo (47.0% of all remaining cases in eastern and central Africa), Nigeria (19.7%), Ethiopia (8.4%), and Cameroon (7.2%) (**Table 3**). In terms of country-specific morbidity prevalence, Gabon has highest overall morbidity prevalence (3.6%; mainly due to high prevalence of reversible skin disease and assumed absence of MDA), followed by the Democratic Republic of Congo (2.9%), and the Republic of Congo (2.8%), and South Sudan (2.3%). There are also some hyperendemic foci with high overall morbidity prevalence (>7%) remaining in the Democratic Republic of Congo and South Sudan by 2030 (mainly due to palpable nodules). See **S3 Text** and **S4 Text** for further detailed estimates by year, project, and country.

## DALYs lost due to onchocerciasis

The total number of DALYs lost slightly increased between 1990 and 2000 due to population growth combined with the gradual roll-out of MDA programmes, but from 2000 onwards we predict a steady decline in DALYs lost (**Fig 1**). We estimate that the total DALYs lost due to onchocerciasis was 2.5 million before MDA started (1990), that this figure declined to 1.4 million in 2020, and will further decline to 691 thousand by 2030. In 1990, most of the onchocercal disease burden was due to blindness (1.4 million DALYs) and severe itch (747 thousand DALYs) (**Table 1**). By 2030, blindness (491 thousand DALYs) and severe itch (98 thousand DALYs), will account for 71.1% and 14.2% of all DALYs lost, respectively (**Table 1**). This

**Table 3. Population at risk and numbers of cases infected and with clinical manifestations by country for 2030.** Absolute numbers (N and DALYs) are presented in thousands.

| Country | Total pop. at risk | Mf infected | | Palpable nodules | | | Reversible skin disease | | | Irreversible skin disease | | | Onchocercal eye disease | | |
|---|---|---|---|---|---|---|---|---|---|---|---|---|---|---|---|
| | | N | % | N | % | DALYs | N | % | DALYs | N | % | DALYs | N | % | DALYs |
| Angola | 3,925 | 101 | 2.6 | 15 | 0.4 | <0.5 | 10 | 0.2 | 2 | 25 | 0.6 | 1 | <0.5 | <0.05 | <0.5 |
| Burundi | 3,944 | 163 | 4.1 | 25 | 0.6 | <0.5 | 17 | 0.4 | 3 | 24 | 0.6 | 1 | <0.5 | <0.05 | <0.5 |
| Cameroon | 15,156 | 426 | 2.8 | 76 | 0.5 | 1 | 46 | 0.3 | 7 | 161 | 1.1 | 7 | 21 | 0.1 | 28 |
| CAR | 3,896 | 62 | 1.6 | 15 | 0.4 | <0.5 | 7 | 0.2 | 1 | 23 | 0.6 | 1 | 20 | 0.5 | 20 |
| Chad | 3,780 | 85 | 2.3 | 18 | 0.5 | <0.5 | 9 | 0.2 | 1 | 15 | 0.4 | <0.5 | 15 | 0.4 | 17 |
| Congo | 2,350 | 144 | 6.1 | 29 | 1.3 | <0.5 | 18 | 0.8 | 3 | 18 | 0.8 | 1 | <0.5 | <0.05 | 1 |
| Democratic Republic of Congo | 68,096 | 3,094 | 4.5 | 795 | 1.2 | 8 | 296 | 0.4 | 46 | 860 | 1.3 | 38 | 35 | 0.1 | 94 |
| Equatorial Guinea | 588 | <0.5 | <0.05 | <0.5 | <0.05 | <0.5 | <0.5 | <0.05 | <0.5 | 3 | 0.5 | <0.5 | <0.5 | <0.05 | <0.5 |
| Ethiopia | 19,911 | 681 | 3.4 | 126 | 0.6 | 1 | 64 | 0.3 | 10 | 163 | 0.8 | 6 | 2 | <0.05 | 4 |
| Gabon | 146 | 17 | 11.9 | 2 | 1.5 | <0.5 | 2 | 1.2 | <0.5 | 1 | 0.9 | <0.5 | <0.5 | <0.05 | <0.5 |
| Liberia | 2,606 | <0.5 | <0.05 | <0.5 | <0.05 | <0.5 | <0.5 | <0.05 | <0.5 | 8 | 0.3 | <0.5 | <0.5 | <0.05 | <0.5 |
| Malawi | 3,469 | <0.5 | <0.05 | <0.5 | <0.05 | <0.5 | <0.5 | <0.05 | <0.5 | 8 | 0.2 | <0.5 | <0.5 | <0.05 | <0.5 |
| Mozambique | 107 | 6 | 5.7 | 1 | 0.6 | <0.5 | 1 | 0.5 | <0.5 | 1 | 0.7 | <0.5 | <0.5 | <0.05 | <0.5 |
| Nigeria | 83,829 | 1,475 | 1.8 | 187 | 0.2 | 2 | 149 | 0.2 | 23 | 298 | 0.4 | 10 | 198 | 0.2 | 207 |
| South Sudan | 11,393 | 323 | 2.8 | 81 | 0.7 | 1 | 28 | 0.2 | 4 | 74 | 0.7 | 3 | 83 | 0.7 | 118 |
| Sudan | 1,083 | 48 | 4.5 | 7 | 0.6 | <0.5 | 5 | 0.5 | 1 | 6 | 0.5 | <0.5 | 7 | 0.6 | 8 |
| Tanzania | 5,444 | 186 | 3.4 | 35 | 0.6 | <0.5 | 19 | 0.3 | 3 | 42 | 0.8 | 2 | 1 | <0.05 | 2 |
| Uganda | 6,379 | <0.5 | <0.05 | <0.5 | <0.05 | <0.5 | <0.5 | <0.05 | <0.5 | 27 | 0.4 | 1 | 1 | <0.05 | 1 |
| **Total** | **236,102** | **6,813** | **2.9** | **1,412** | **0.6** | **15** | **670** | **0.3** | **104** | **1,756** | **0.7** | **71** | **384** | **0.2** | **501** |

amounts to a reduction in DALYs lost due to blindness and severe itch of 64.9% and 86.9%, respectively. Nigeria, the Democratic Republic of Congo, and South Sudan will carry the highest disease burden due to onchocerciasis in 2030 (Table 3 and Fig 2), primarily because of the high number of DALYs lost due to blindness in savanna areas and the high numbers of people living in endemic areas in those countries. Overall, thanks to MDA, the annual number of DALYs averted will rise to 6.6 million by 2030 (Fig 1) and around 96.7 million DALYs will have been cumulatively averted between 1990 and 2030.

## Sensitivity analysis

Fig 3 shows the impact of alternative assumptions about the estimated number of cases with reversible and irreversible OSD by 2030. For reversible skin manifestations, the assumptions regarding MDA implementation have the highest impact on case estimates, with up to 826.8 thousand more estimated cases when MDA coverage was systematically reported to be higher (20% points) than the actual distribution. As *O. volvulus* infection is directly related to the presence of acute, reversible skin conditions, earlier initiation of MDA in untreated areas would reduce case estimates by 117.7 thousand cases. For chronic, irreversible skin manifestations, the effect of systematic over-reporting of MDA coverage would be less pronounced on current case estimates (up to 316.5 thousand more cases if MDA coverage is over-reported by 20% points). Halving the factor with which we calculate morbidity prevalence in hypoendemic as a function of prevalence in mesoendemic areas would considerably influence our case estimates with irreversible subtypes of OSD for 2030 (545.4 thousand fewer cases). As for OSD, the predicted prevalence of skin mf and nodules in 2030 was most sensitive to assumptions about MDA coverage. In contrast, OED estimates for 2030 were only marginally sensitive to MDA coverage but were

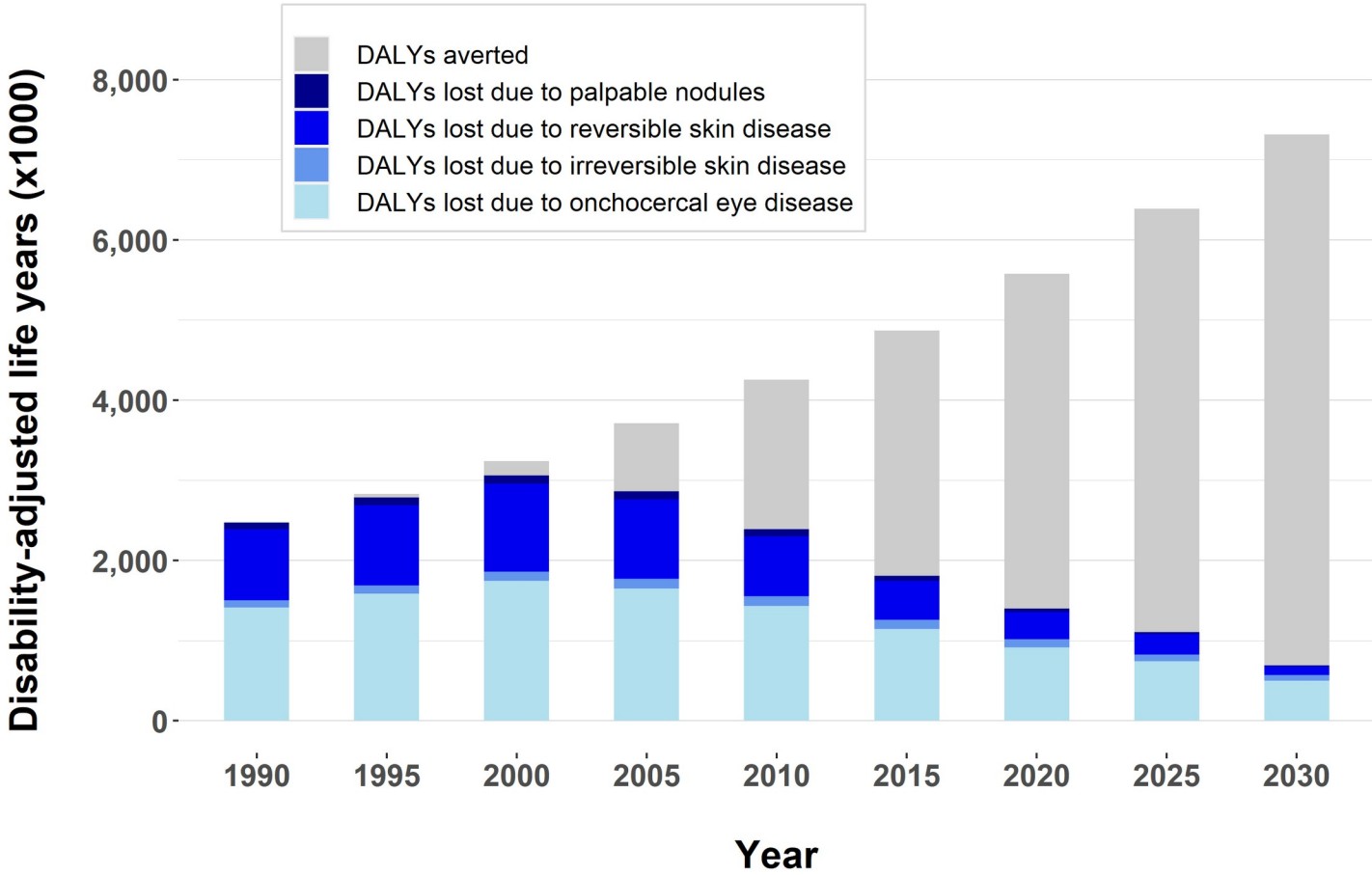

**Fig 1. Total number of disability-adjusted life years (DALYs) lost and averted due to onchocerciasis from 1990 to 2030 in countries formerly covered by the African Programme for Onchocerciasis Control.** DALYs are expressed in thousands (x1000). The total height of the blue part of the bars represents the estimated total number of DALYs lost, with different intensities of blue representing four subcategories of onchocercal morbidity (see legend). The total height of the bars (blue plus grey) represents the DALYs lost in a counterfactual scenario without large-scale MDA. The light grey part of each bar therefore represents the annual number of DALYs averted by MDA.

highly sensitive to assumptions about excess mortality due to blindness and potential reversibility of visual impairment, which led to up to an estimated 7.2% and 93.3% fewer cases of OED respectively by 2030 (**S1 Text**, section 5). Starting MDA in as yet untreated *L. Loa*-endemic areas in 2024 instead of 2025, and in untreated non-*L. loa*-endemic areas in 2022 instead of 2023 would considerably reduce the number of cases with reversible skin manifestations by 2030 (-17.6%), but not the number of cases with irreversible skin morbidity (-1.1%).

## Discussion

We predict major reductions in the burden of onchocerciasis between 1990 and 2030, both in terms of number of cases as well as in DALYs lost, mainly thanks to the massive impact of MDA with ivermectin. We predict that the annual number of DALYs lost due to onchocerciasis will be more than halved over a 40-year time frame, with around 97 million DALYs cumulatively averted by MDA between 1990 and 2030 in countries formerly covered by APOC. Most of the cases remaining by 2030 will be due to palpable nodules and chronic skin manifestations, and most of the remaining burden will be due to severe itch and blindness. By 2030,

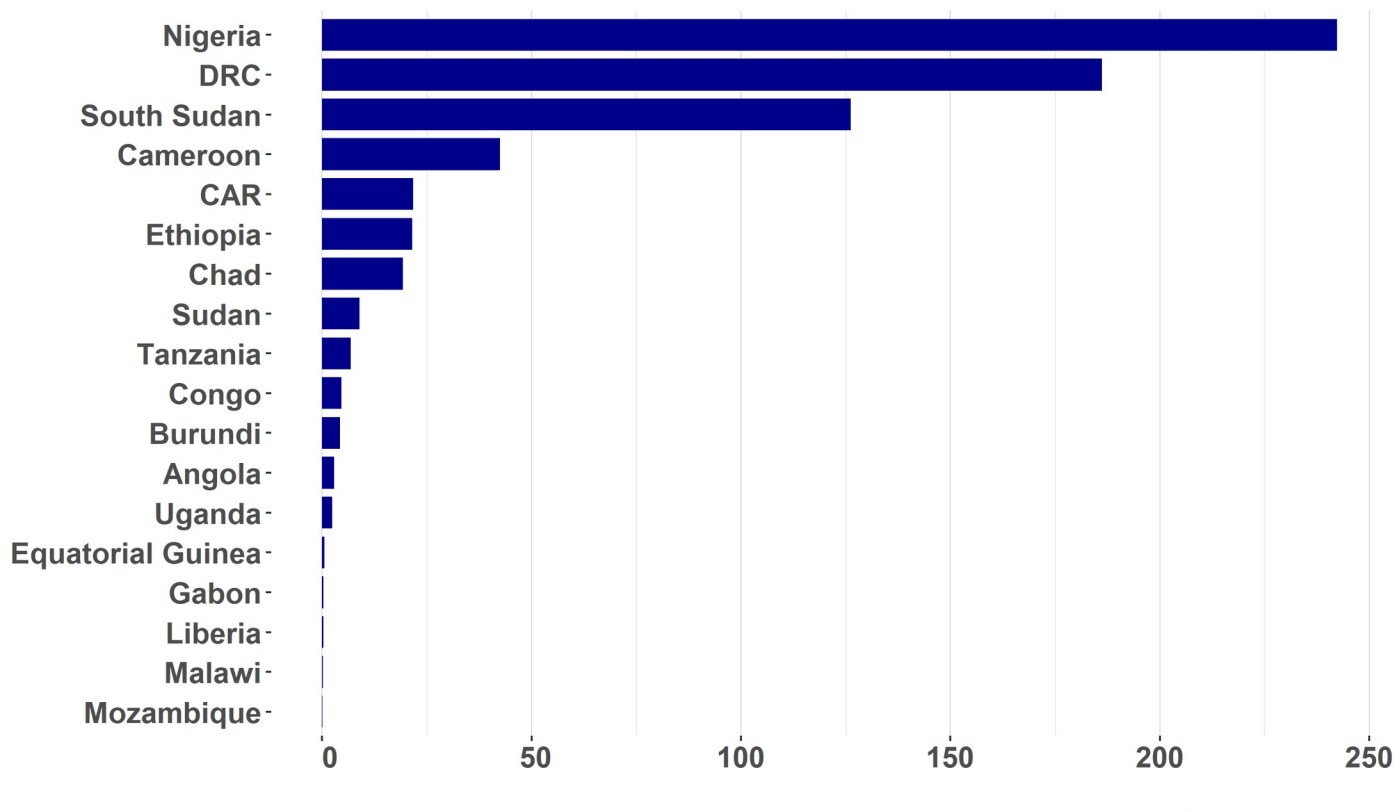

**Fig 2. Total number of disability-adjusted life years (DALYs) lost due to onchocerciasis by country in 2030.**

almost half of all remaining cases will be located in the Democratic Republic of Congo, where many people live in endemic areas and MDA started relatively late.

The GBD study estimated that the number of DALYs lost due to onchocerciasis was 1.4 million in 1990 and 1.3 million in 2015 [32,33]. The difference with our burden estimates for the same years (2.5 million and 1.8 million DALYs lost in 1990 and 2015, respectively) can be largely explained by the fact that excess mortality due to onchocercal blindness was not considered in the GBD study, while we estimated onchocercal blindness to be responsible for 1.4 million and 1.1 million DALYS lost in 1990 and 2015, respectively. The remaining difference may be explained by a methodological difference: the GBD study relies on statistical models while our study uses a mechanistic mathematical model, which we consider more appropriate for producing forecasts.

We identified areas where we predict high remaining onchocerciasis cases (due to loiasis co-endemicity or very hyperendemic baseline infection levels), as well as areas where onchocerciasis transmission may be close to elimination [5]. Our results are in line with a recent analysis on remaining *O. volvulus* and *L. loa* cases by 2025 [14], predicting 19.0% *O. volvulus* and 5.8% *L. loa* mf prevalence in these areas. This highlights that particularly hypoendemic areas with loiasis transmission, currently excluded from mass treatment, should rapidly be targeted for onchocerciases elimination mapping and implementation of alternative treatment strategies, such as the Test-and-not-Treat strategy, community-directed vector control (e.g. slash and clear vegetation, ground-based larviciding), or possibly a new macrofilaricidal

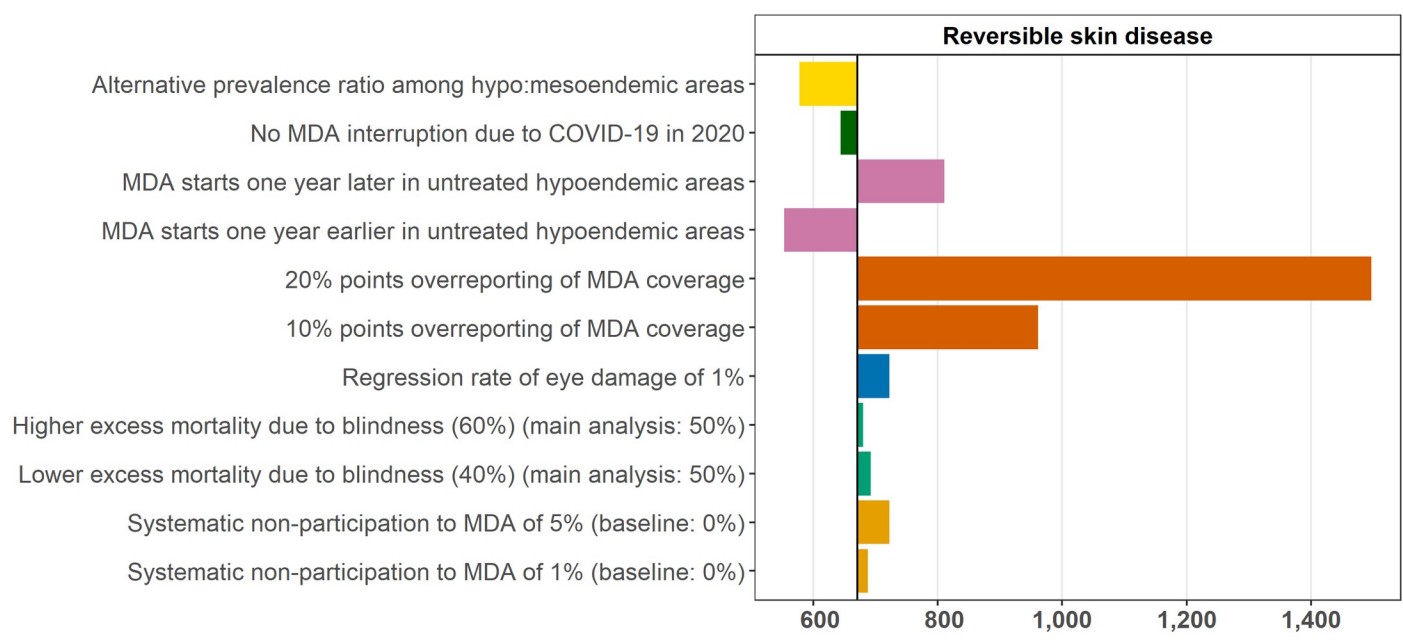

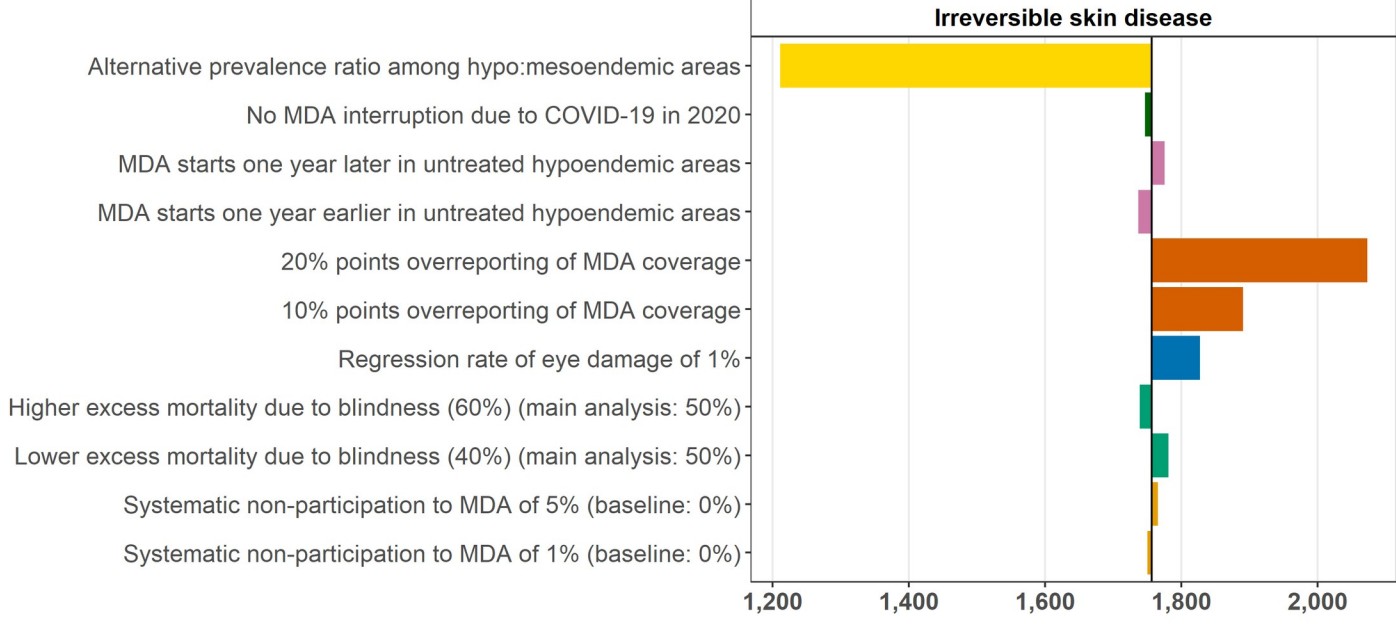

**Fig 3. Univariate sensitivity analysis for the predicted number of cases with reversible and irreversible onchocercal skin disease by 2030.** Coloured bars represent the difference between the sensitivity analysis and the results of the main analysis (vertical black line). * In the sensitivity analysis, the prevalence of infection and clinical manifestations in hypoendemic areas was assumed to be 1/3 of that in mesoendemic areas (as applied in a previous modelling exercise [2]), instead of ratios based on a more detailed meta-analysis performed for the main analysis (see also **Table E** in **S1 Text**).

treatment for onchocerciasis, once available. Intensified interventions could also be considered to bring down infection and morbidity in areas with historically high baseline endemicity levels that are currently still hyperendemic, even after long-term control. We predict that by 2030

the highest remaining prevalence of infection and morbidity is likely to be found in endemic areas of Gabon, Democratic Republic of Congo, and the Republic of Congo. However, the highest burden will be found in countries with the highest number of population at risk or those with the blinding savanna type of onchocerciasis (i.e. Nigeria, DRC, South Sudan, Cameroon). In other countries, the long-standing control interventions in Africa will have reduced infection and morbidity to negligible levels, such as across large areas in Burundi, Equatorial Guinea, and Uganda. For Uganda, these predictions are in line with empirical evidence: 15 out of 17 onchocerciasis-endemic foci in Uganda are currently under post-treatment surveillance or have eliminated onchocerciasis [34,35].

The presented number of cases with infection and morbidity should be considered as indicative only, as the estimates depend on uncertain assumptions regarding baseline endemicity levels, local transmission conditions and programmatic factors. Numbers presented for any hypo-endemic areas left untreated by APOC should be considered with particular caution. The P5 areas include all areas with an estimated nodule prevalence between 5% and 20%, based on a geostatistical analysis of REMO survey data [11]. However, in most areas hypo-endemicity remains to be confirmed in the field through onchocerciasis elimination mapping, although the ESPEN website suggest that elimination mapping may not, or no longer be required in seven former APOC-countries (i.e., Burundi, Chad, Equatorial Guinea, Liberia, Malawi, Tanzania and Uganda). In some of the P5 areas, onchocerciasis infection prevalences may have been reduced thanks to MDA programmes against lymphatic filariasis in loiasis-free areas or the collateral benefits of other drug programmes (e.g., ivermectin against scabies in Ethiopia [36]) may have led to minor overestimations of the aforementioned burden estimates for these areas.

Although we have taken account of most clinical manifestations attributable to *O. volvulus* infection, there is increasing evidence of an association between high intensity *O. volvulus* infection and the development of epilepsy during childhood [37–40]. A recent study reported 128 thousand YLDs by 2015 due to onchocerciasis-associated epilepsy in 9 of the 17 countries formerly covered by APOC [41]. This would imply an additional 53.8 thousand DALYs lost due to onchocerciasis (128,000 cases x 0.420 disability weight for uncontrolled epilepsy [25]) by 2015, which is a conservative estimate as it does not capture YLLs caused by excess mortality associated with epilepsy in rural Africa [41]. In addition, MDA with ivermectin will also have a modest impact on off-target diseases like soil-transmitted helminths and scabies [42].

The results presented here are estimates based on the most comprehensive available data from countries formerly covered by APOC. The study was motivated by the extreme paucity of large-scale representative data on actual morbidity levels in (formerly) onchocerciasis-endemic areas. Disease manifestations were modelled using a newly developed disease module within ONCHOSIM that, for the first time, could simultaneously simulate reversible and irreversible clinical manifestations, as well as single- and multi-stage disease, taking into account the impact of excess mortality due to blindness on the trends for prevalence of all these conditions. Model outcomes were externally validated against longitudinal trends in morbidity prevalence, community-level patterns in prevalence of infection and disease manifestations, age-stratified pre-control morbidity prevalence data from savanna communities, and data on the concurrence of clinical manifestations, and it was shown that the model could reasonably well reproduce the morbidity patterns in the field [7]. Also, it is important to note that our analysis builds on estimates of the pre-control distribution of infection levels based on REMO (rapid epidemiological mapping of onchocerciasis) data, which are the most comprehensive and accurate data available, and the basis of the current map of onchocerciasis in the 20 APOC countries. The oldest REMO surveys possibly date from 1993, but many were done more recently. These REMO data provide a good indication of pre-control prevalence in treated

areas. The extent of hypoendemic areas is more uncertain; unfortunately, data from recent and ongoing mapping efforts in hypoendemic areas with more sensitive diagnostic tools are not yet published. We further note that we did not capture the impact of potential secular developments, such as climate change, deforestation, economic development, demographic transition, and urbanisation, which may lead or may have led to changes in transmission intensity and lower population densities in endemic areas than assumed here. However, we do not expect this to impact our results with regard to where most of the future onchocercal burden will be. Indeed, the geographical distribution of infection may have changed somewhat over time prior to start of MDA. However, as we aggregated results over a larger geographical area, we expect that this uncertainty does not affect, or only marginally affects, our main findings and conclusions.

We used the mathematical model ONCHOSIM to simulate the impact of ivermectin in treated areas on infection and morbidity according to reported treatment history and future MDA scenarios. ONCHOSIM captures community infection dynamics throughout MDA quite well [2,10,43,44], but relies on adequate input about MDA coverage levels, which may be reported imprecisely or incorrectly [45]. Our sensitivity analysis suggests that a 10%- or 20%-point change (everywhere) in coverage of MDA has a considerable impact on case and burden estimates, highlighting the importance of high-quality MDA coverage data. We based the assumptions of control interventions on information from the APOC treatment database [15] and the ESPEN portal. We might be unaware of acceleration strategies that countries may have implemented in specific implementation units, such as changing MDA frequency from annual to bi-annual [35,46]. Also, apart from (community-based) vector control on the island of Bioko [47–50] and in Uganda [51–53], we did not take account of any other local vector control strategies that may have been implemented in other countries. However, we expect that in general, local interventions other than annual MDA would only have a minimal additional impact on the burden of disease, given the already large effect of annual MDA with ivermectin on prevalence of acute clinical manifestations and incidence of chronic conditions. We have further taken account of major events that may have influenced the continuation of the MDA programme such as civil war in South Sudan, the CAR and Liberia, as well as the COVID-19 pandemic. The WHO recommended in 2020 to suspend all epidemiological surveys and MDA activities for Neglected Tropical Diseases tackled by preventive chemotherapy and transmission control due to the COVID-19 pandemic [54]. A recent modelling paper assessed the impact of MDA disruptions and delays in scaling-up of MDA since 2020, and shows that this affects *O. volvulus* transmission and infection level in both the short and long term [55].

We used the latest GBD disability weights [23], mapping the clinical manifestations of onchocerciasis to the lay descriptions with corresponding disability weights to estimate DALYs lost due to onchocerciasis. The disability weight for blindness was estimated at 0.187 by the GBD study. Several previous estimates of the blindness disability weight were considerably higher [56]. The difference between earlier and current estimates may be primarily due to the dimensions covered by the disability weight: the GBD disability weights focus only on health loss due to blindness, not capturing other socio-economic consequences of blindness that may occur in Africa (e.g., related to livelihood, employment, accessibility of public resources and infrastructure). Such socio-economic consequences are therefore also not captured in our burden estimates. Quantifying this socio-economic burden was beyond the scope of this work, but it is important to acknowledge that this burden will be high, especially for blindness in the African context [57–59].

Our analysis only covers countries in Central and East Africa formerly covered by APOC, whereas in West Africa the Onchocerciasis Control Programme (OCP) also has been

recognised as one of the most successful programmes in the history of development aid [35]. Therefore, a next step is to perform a similar modelling study for countries previously under the OCP mandate to obtain a comprehensive synopsis of health losses due to onchocerciasis in the whole of Africa. A major challenge here will be to collate data on the history of MDA and vector control, which, unlike the case for APOC countries, was not curated centrally but by the individual countries that were part of OCP.

## Conclusions

MDA has had a remarkable impact on the onchocerciasis burden in countries previously under the APOC mandate. Yet, by 2030 we still expect over 10 million mf-positive people and almost 20 million individuals harbouring adult worms, with the majority living in only a few countries. If in the future elimination of onchocerciasis is achieved, acute clinical manifestations of onchocerciasis such as reactive skin disease and severe itch will have largely disappeared, although chronic clinical manifestations will linger and only slowly disappear due to demographic turn-over. In the few countries where we predict continued transmission between now and 2030, intensified MDA could be combined with local vector control efforts, or the introduction of new drugs for mopping up residual cases of infection and morbidity.

## Supporting information

**S1 Text. A PDF file with a detailed description of the methodology and additional details of sensitivity analyses.**
(PDF)

**S2 Text. A PDF file with treatment history and assumptions per APOC project used in simulations.**
(PDF)

**S3 Text. A PDF file with additional tables with estimated case numbers by age, sex, endemicity level, and APOC projects for 1990, 2020, and 2030.**
(PDF)

**S4 Text. A PDF file with detailed estimates (number of cases, and DALYs) by country and APOC project.**
(PDF)

## Acknowledgments

We would like to warmly thank Louise Burrows (DND*i*) for editing the final draft of the manuscript. We would like to acknowledge WHO/AFRO for approving the use and publication of the data used in this manuscript. We are grateful to Hans Remme for helping in compiling pixel-level mf prevalence data and stratifying these pixels per APOC project over endemicity levels and coupling to population census data. In addition, we would like to thank Federica Giardina for their statistical advice in the programming of R during the initial modelling and analysis stage.

## Author Contributions

**Conceptualization:** Wilma A. Stolk, Belén Pedrique, Luc E. Coffeng.

**Data curation:** Natalie V. S. Vinkeles Melchers, Welmoed van Loon.

**Formal analysis:** Natalie V. S. Vinkeles Melchers, Luc E. Coffeng.

**Funding acquisition:** Luc E. Coffeng.

**Investigation:** Natalie V. S. Vinkeles Melchers, Wilma A. Stolk, Sake J. de Vlas, Luc E. Coffeng.

**Methodology:** Wilma A. Stolk, Michele E. Murdoch, Sake J. de Vlas, Luc E. Coffeng.

**Project administration:** Luc E. Coffeng.

**Software:** Roel Bakker.

**Supervision:** Wilma A. Stolk, Sake J. de Vlas, Luc E. Coffeng.

**Visualization:** Natalie V. S. Vinkeles Melchers, Welmoed van Loon.

**Writing – original draft:** Natalie V. S. Vinkeles Melchers.

**Writing – review & editing:** Wilma A. Stolk, Welmoed van Loon, Belén Pedrique, Roel Bakker, Michele E. Murdoch, Sake J. de Vlas, Luc E. Coffeng.

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
