## [Decision Letter · Decision Letter 0]

26 Dec 2020

Dear Dr. Coffeng,

Thank you very much for submitting your manuscript "The burden of skin disease and eye disease due to onchocerciasis in Africa for 1990, 2015, and 2025" for consideration at PLOS Neglected Tropical Diseases. As with all papers reviewed by the journal, your manuscript was reviewed by members of the editorial board and by several independent reviewers. In light of the reviews (below this email), we would like to invite the resubmission of a significantly-revised version that takes into account the reviewers' comments. 

We cannot make any decision about publication until we have seen the revised manuscript and your response to the reviewers' comments. Your revised manuscript is also likely to be sent to reviewers for further evaluation.

Sincerely,

Alberto Novaes Ramos Jr

Guest Editor

Sara Lustigman

Deputy Editor

Reviewer's Responses to Questions

**Key Review Criteria Required for Acceptance?**

**Methods**

-Are the objectives of the study clearly articulated with a clear testable hypothesis stated?

-Is the study design appropriate to address the stated objectives?

-Is the population clearly described and appropriate for the hypothesis being tested?

-Is the sample size sufficient to ensure adequate power to address the hypothesis being tested?

-Were correct statistical analysis used to support conclusions?

-Are there concerns about ethical or regulatory requirements being met?

Reviewer #1: -Are the objectives of the study clearly articulated with a clear testable hypothesis stated? Yes 

-Is the study design appropriate to address the stated objectives? Yes 

-Is the population clearly described and appropriate for the hypothesis being tested? NAP

-Is the sample size sufficient to ensure adequate power to address the hypothesis being tested? NAP

-Were correct statistical analysis used to support conclusions? yes

-Are there concerns about ethical or regulatory requirements being met No

Reviewer #2: The objectives are clearly elucidated but the process is very theoretical and although well explained it is unrelated to the reality in the field in East and Central Africa.

Reviewer #3: (No Response)

Reviewer #4: This is a robust analysis using a well established method. The objectives of the study are clearly stipulated. The design of the analysis is appropriate to answer the objectives of the study. One thing the authors should clarify is that the countries supported by APOC are less than the currently endemic countries. So this should be clarified so that the figures here not to look like for Africa.

**Results**

-Does the analysis presented match the analysis plan?

-Are the results clearly and completely presented?

-Are the figures (Tables, Images) of sufficient quality for clarity?

Reviewer #1: -Does the analysis presented match the analysis plan? yes 

-Are the results clearly and completely presented? Yes 

-Are the figures (Tables, Images) of sufficient quality for clarity? Sufficient

Reviewer #2: The analysis presented do match the plan but there are flaws in the data sources and therefore the conclusions. The figures are clear.

Reviewer #3: (No Response)

Reviewer #4: The results are clear.

**Conclusions**

-Are the conclusions supported by the data presented?

-Are the limitations of analysis clearly described?

-Do the authors discuss how these data can be helpful to advance our understanding of the topic under study?

-Is public health relevance addressed?

Reviewer #1: -Are the conclusions supported by the data presented? yes 

-Are the limitations of analysis clearly described? yes 

-Do the authors discuss how these data can be helpful to advance our understanding of the topic under study? yes

-Is public health relevance addressed? yes

Reviewer #2: Conclusions are supported by the data presented but these are so general that this study does not really add to what is already known.

Reviewer #3: (No Response)

Reviewer #4: (No Response)

**Editorial and Data Presentation Modifications?**

Reviewer #1: see comments below.

Reviewer #2: See comments below.

Reviewer #3: 1. The title of the paper seems to imply that this paper will provide results for the whole of Africa, and should be changed to make it clear that it only covers (some) ex-APOC countries.

2. The authors might like to explain why they did not include all the ex-APOC countries (i.e. Liberia and Kenya), and make it clear that they did not cover all ex-APOC countries in the abstract.

3. The authors seem to put more emphasis on macrofilaricides as a solution to annual MDA effectiveness issues, as opposed to other interventions. They do mention other possibilities, but it seems that the authors are suggesting that macrofilaricides would be the solution of choice, or possibly the only solution in some cases (e.g. lines 293 and 284). Please can the authors explain this, or make some small revisions so that the solutions are less one-sided.

Reviewer #4: None.

**Summary and General Comments**

Reviewer #1: Excellently written paper, very extensive, well documented modelling work. 

However as with all modelling work results depend on the available data and these are relatively poor. 

First the model relies on REMO data, collected in 1993. Since 1993 a lot of changes other than onchocerciasis elimination efforts may have changed the onchocerciasis prevalence in sub-Saharan Africa. 

Second, the model relies on ivermectin coverage as reported by the onchocerciasis control/elimination programs. These coverage rates underestimate in many areas, mainly in those with most onchocerciasis related morbidity the true coverage. 

Bi-annual CDTI and vector control was not taking into account because as the authors state “it was considered this would not make a big difference given the already large effect of annual MDA with ivermectin on prevalence of acute clinical manifestations and incidence of chronic conditions.” I can see it is difficult to take this into account. However we have seen that implementing bi-annual CDTI in combination with vector control made a major difference concerning onchocerciasis related morbidity (epilepsy) in northern Uganda (1). We hope that a similar approach would be implemented in other areas of high O volvulus transmission and very high onchocerciasis morbidity for example in South Sudan. 

1. N. Gumisiriza et al., Prevalence and incidence of nodding syndrome and other forms of epilepsy in onchocerciasis-endemic areas in northern Uganda after the implementation of onchocerciasis control measures. Infect Dis Poverty 9, 12 (2020).

The authors mention they did not capture “the impact of potential secular developments such as deforestation, economic development, demographic transition, and urbanisation, which may have led to changes in transmission intensity and a lower (future) population at risk living in endemic areas than assumed here.” 

They do not mention however security problems, occurring frequently in certain African countries, who may seriously complicate the distribution of ivermectin. The current COVID-19 epidemic is another example that has interrupted onchocerciasis elimination efforts. 

It is good the authors mention the association between high intensity O. volvulus infection and the development of epilepsy during childhood [28–31]. However they mention this important public health problem in onchocerciasis endemic regions only in the discussion. I propose to mention this already in the introduction. Indeed epilepsy is an onchocerciasis associated condition with very high disability, much higher than skin lesions and blindness. 

The authors predict that the highest country-level mf prevalence in 2025 is expected in Gabon (19.4%), where currently no MDA programme is in place due to Loa loa co-endemicity, Mozambique (11.7%), Republic of Congo (9.5%), and the Democratic Republic of Congo (8.2%) (Table 3). I’m surprised that countries that experience war with complete disruption of CDTI programs such as South Sudan and CAR are not mentioned. 

The authors state “Disability weights are based on a previously published multi-country study by Salomon et al.[31]

This study by Soloman analysed data from new web-based surveys of participants aged 18-65 years, completed in four European countries. These data are clearly not relevant for the O. volvulus infected person. 

For example a score is given for “Disfigurement: level 2. Person has a visible physical deformity that causes others to stare and comment. As a result, the person is worried and has trouble sleeping and concentrating.” 

I have difficulties to believe that chronic irreversible skin problems in O. volvulus infected persons are associated with more disability than O. volvulus related serious vision problems.

Reviewer #2: General Comments to authors

I congratulate the authors on the enormous and detailed work that you have done on this paper, but with all the assumptions (that you have well explained), I wonder if it would not have been better to look at what is happening on the ground and to look at more impact data. There is too much emphasis on coverage data much of which is unrealistic. I feel fo all the work done your conclusions are in fact very weak.

Specific comments to authors

Line 76. Agreed; 217.5 million people needing PC but to be more realistic you should add how many are nearing the end of treatment.

Line 86: You mention people with disease but you need to make clearer the difference between transmission of the disease and manifestations of disease (morbidity) in the text.

Line 173. Some hypoendemic foci may be driven by spill over from hyperendemic foci but by no means all of them. Look at Gabon!

Line 189. You explain the modifications to GBD for the skin manifestations of onchocerciasis. A similar approach must be made to visual impairment and blindness as once again the GBD does not really take into account the impact of blindness in an onchocerciasis environment.

Line 228. OED was hardly measured in the APOC region. Most of the surveys done by APOC in the early days remained unpublished and were not followed up.

Line 246. I think you may need to add South Sudan to your list. Surveys in the APOC era showed the results reported were not realistic, and there has been virtually no treatment for a period of about 10 years until 2017 when treatment has started to scale up again. There are undoubtedly new cases of blindness and there is a lot of epilepsy in the south which you have mentioned but will be a serious problem for morbidity management later.

Line 298. I agree that excess mortality due to OED needs to be taken into account but there is definitely a difference between blindness trends due to vector control and those due to ivermectin. With ivermectin use more early cases are reversible where these are not impacted early enough with vector control. 

Line 350. In these days of integration of PC it is unrealistic to not take into account treatment s for LF, particularly in hypoendemic areas.

Line 384. I agree with your conclusion that macrofilaricidal drugs will be needed to mop up hot spots and treating clinical cases and the sequelae of onchocerciasis may well continue after elimination of transmission will have been achieved.

Figure 1 DALYs lost due to skin nodules???

Figure 1 and 3. Clinical signs of irreversible skin disease remain, I agree. I am not sure how much these really impact daily life once under treatment with ivermectin. There are some social consequences but I am not sure how you arrive at these DALYs

Supplementary material.

S2 No mention of Loa loa in DRC. It is the worst affected country and will play a major role in the treatment of hypoendemic areas.

S2 and S4. I know assumtions have to be made but having straight line coverages between 75 and 80% are not at all realistic, especially in unstable countries which as you say is where most of the burden will be in the future.

Just a few specific points on the country profiles.

Malawi is only working on cross border issues with Mozambique

3 States in Nigeria have already stopped treatment yet you graph shows treatment continuing

South Sudan. There was virtually no treatment for up to 10 years and even an APOC survey found the reported coverage to be inadequate. Treatment started again in 207 but is still only just scaling up to 100% coverage. There is a lot of morbidity as already mentioned. 

Uganda. Treatment has already stopped in all foci apart from 1 in the North where the LRA was active.

Burundi. Most surveys have been negative. There is only the problem of security before doing impact surveys.

Reviewer #3: This is a good and useful paper which deserves publication. It is well written, easy to read and needs very little revision. However, please note that if the other (accompanying) paper submitted by the authors (which I have not been asked to review) is rejected or requires major revision, that may have consequences for this paper.

Reviewer #4: The analysis by Vinkeles Melchers and colleagues on the burden of skin disease and eye disease due to onchocerciasis in Africa for 1990, 2015, and 2025 is very important. The estimates provide the burden of skin and eye diseases due to onchocerciasis what progress has been achieved and what efforts are required to achieve the 2025 targets and the 2030 NTD Roadmap targets. I have the following comments though. 

• The title says Africa in fact the analysis was focusing only in the APOC countries. This should come out in the title as well.

• The authors should clarify on the list of APOC supported countries and what is the difference between the current scope of the onchocerciasis programme compared with the APOC. The authors have the liberty to analyse the data for APOC supported countries, but they should make clear distinction that there are additional countries which were not covered by APOC now in the elimination programme in Africa.

PLOS authors have the option to publish the peer review history of their article (what does this mean?). If published, this will include your full peer review and any attached files.

Reviewer #1: Yes: Robert Colebunders

Reviewer #2: No

Reviewer #3: No

Reviewer #4: No
---

## [Decision Letter · Decision Letter 1]

29 Jun 2021

Dear Dr. Coffeng,

We are pleased to inform you that your manuscript 'The burden of skin disease and eye disease due to onchocerciasis in countries formerly under the African Programme for Onchocerciasis Control mandate for 1990, 2020, and 2030' has been provisionally accepted for publication in PLOS Neglected Tropical Diseases.

Best regards,

Alberto Novaes Ramos Jr

Associate Editor

Sara Lustigman

Deputy Editor

Reviewer's Responses to Questions

**Key Review Criteria Required for Acceptance?**

**Methods**

-Are the objectives of the study clearly articulated with a clear testable hypothesis stated?

-Is the study design appropriate to address the stated objectives?

-Is the population clearly described and appropriate for the hypothesis being tested?

-Is the sample size sufficient to ensure adequate power to address the hypothesis being tested?

-Were correct statistical analysis used to support conclusions?

-Are there concerns about ethical or regulatory requirements being met?

Reviewer #1: The authors responded appropriately to the comments of the reviewers and adapted the paper accordingly

Reviewer #2: -Are the objectives of the study clearly articulated with a clear testable hypothesis stated? YES

-Is the study design appropriate to address the stated objectives? YES

-Is the population clearly described and appropriate for the hypothesis being tested? YES

-Is the sample size sufficient to ensure adequate power to address the hypothesis being tested? YES

-Were correct statistical analysis used to support conclusions? YES

-Are there concerns about ethical or regulatory requirements being met? NO

**Results**

-Does the analysis presented match the analysis plan?

-Are the results clearly and completely presented?

-Are the figures (Tables, Images) of sufficient quality for clarity?

Reviewer #1: The authors responded appropriately to the comments of the reviewers and adapted the paper accordingly

Reviewer #2: The results analysis match the analysis plan and are well represented, and are presented clearly

**Conclusions**

-Are the conclusions supported by the data presented?

-Are the limitations of analysis clearly described?

-Do the authors discuss how these data can be helpful to advance our understanding of the topic under study?

-Is public health relevance addressed?

Reviewer #1: The authors responded appropriately to the comments of the reviewers and adapted the paper accordingly

Reviewer #2: The limitations of the data are well described. The conclusions are represented by the data although some of the assumptions are a bit weak. Stigmatisation of people with nodules may occur in some communities but this is rare nothing like the stigmatisation of other skin disease and is overstated. I think this makes the scenario in 2030 appear worse than it probably would be but this has been explained. I think this paper is useful from the public health point of view but it could stress the need for the development alternative strategies to avoid some of the issues in the problem countries.

**Editorial and Data Presentation Modifications?**

Reviewer #1: (No Response)

Reviewer #2: The authors have worked hard to answer the questions raised by the reviewers and have brought the projections up to 2030 which fits in much better with the SDGs and the WHO road map.

It would be good to have some definitions in the text and not in the subsidiary information, particularly the P5 and P20 APOC projects which are not commonly used.

**Summary and General Comments**

Reviewer #1: The authors responded appropriately to the comments of the reviewers and adapted the paper accordingly

Reviewer #2: This paper is much improved and although I disagree with some of the assumptions they have been well explained (and justified). I think it could be a useful paper in long term planning and hopefully will push development for new tools and also for support for the "problem countries.

PLOS authors have the option to publish the peer review history of their article (what does this mean?). If published, this will include your full peer review and any attached files.

Reviewer #1: **Yes: **Robert Colebunders

Reviewer #2: No

---

## [Editor Report · Acceptance letter]

9 Jul 2021

Dear Dr. Coffeng,

We are delighted to inform you that your manuscript, "The burden of skin disease and eye disease due to onchocerciasis in countries formerly under the African Programme for Onchocerciasis Control mandate for 1990, 2020, and 2030," has been formally accepted for publication in PLOS Neglected Tropical Diseases.

Best regards,

Shaden Kamhawi

co-Editor-in-Chief

Paul Brindley

co-Editor-in-Chief
